# Eco-Capabilities as a Pathway to Wellbeing and Sustainability

**Nicola Walshe** [1,*], **Zoe Moula** [1] and **Elsa Lee** [2]

1   IOE, UCL's Institute of Education and Society, University College London, London WC1E 6BT, UK; z.moula@ucl.ac.uk
2   School of Education and Social Care, Anglia Ruskin University, Cambridge CB1 1PT, UK; elsa.lee@aru.ac.uk
*   Correspondence: n.walshe@ucl.ac.uk

**Abstract:** Eco-Capabilities is an AHRC funded project situated at the intersection of three issues: a concern with children's wellbeing; their disconnect with the environment; and a lack of engagement with arts in school curricula. It builds on Amartya Sen's work on human capabilities as a proxy for wellbeing, developing the term eco-capabilities to describe how children define what they feel they need to live a fully good human life through environmental sustainability, social justice and future economic wellbeing. A total of 101 children aged 7–10 from schools in highly deprived areas participated in eight full days of arts in nature practice. The study drew on arts based research methods, participatory observations, interviews and focus groups with artists, teachers and children. Findings suggest that arts in nature practice contributed towards eight (eco-)capabilities: autonomy; bodily integrity and safety; individuality; mental and emotional wellbeing; relationality: human/nonhuman relations; senses and imagination; and spirituality. This was facilitated through four pedagogical elements: extended and repeated arts in nature sessions; embodiment and engaging children affectively through the senses; 'slowliness', which envelops children with time and space to (re)connect; and thoughtful practice, which facilitates emotional expression. We suggest that, through these elements, arts in nature practice supports children's wellbeing, and guides them towards a more entangled relationship with nature and a clearer understanding of themselves as part of it, thereby motivating them to take better care of it.

**Keywords:** eco-capabilities; environmental and sustainability education; wellbeing; arts; nature; artscaping; outdoors; mental health; school; children

## 1. Introduction

Global interest in children's wellbeing is growing and is now central to major international policy documents regarding children's life quality (e.g., UN Sustainable Development Goal 3: Good Health and Wellbeing [1]). Research suggests that children's wellbeing is linked to developing positive learning attitudes and coping successfully with change [2]; conversely, low emotional wellbeing can lead to mental health problems [3]. Critically, one in six children in England suffer a severe mental health illness and suicide is the third leading cause of death in young people [4]. However, this is not uncommon within the 'Global North', with Biddle et al. [5] reporting that Ireland, Portugal, Germany, and Finland have the highest rates of reported depression in Europe for those aged 15 years and over [6]. In Australia, there are 1.2 million mental health related general practice encounters for young people annually, and this has increased by 21% during the 2000s [7]. These figures are higher for vulnerable groups, such as those from low income households, those with special educational needs/neurodevelopmental differences (SEN/ND), or those who have been exposed to adverse childhood experiences [8]. There is also increasing evidence that climate change and the ecological crisis is further impacting the mental health and wellbeing of children and young people through eco-anxiety [9,10]. Worryingly, 70% of children and adolescents who experience mental health problems in the UK have not received appropriate support at a sufficiently early age [11], yet the National Service Framework

emphasised for those children at risk of developing mental health problems, assessment of their needs and provision of early intervention can make a significant difference [12].

Substantial benefits for wellbeing may be derived from contact with nature, and lack thereof in childhood has been found to be a predictor for adulthood depression. Recent studies of the impact of COVID-19 on children's mental health have found that regular time outdoors is associated with better mental health, whereas increases in daily screen-time and less time outdoors were associated with poorer mental health [13,14]. Further, in England, the People and Nature Survey found that, while COVID-19 had an impact on children's mental health, children who reported spending more time outside (and more time noticing nature/wildlife) were more likely to report that 'being in nature makes me very happy' (91% and 94%, respectively, compared to 79% of those who had spent less time) [15]. However, six in ten children (60%) reported having spent less time outdoors since the start of coronavirus and the initial lockdown. Consequently, in the UK, US and other 'Global North' societies, there is concern about children's reduced outdoor experience and consequent loss of connection with the natural environment, e.g., [16,17]. In some cases, this has led to the prescription of nature based health interventions, or green prescribing, e.g., [18]; although evidence for its use is currently predominantly limited to adults, it is a convincing indicator of the strength of the impact of this extinction of experience.

Within the UK, HM Government's 'A Green Future: Our 25 Year Plan to Improve the Environment' [19] explicitly commits to helping people improve their health and wellbeing by using green spaces: "we will scope out how we could connect people more systematically with green space to improve mental health" (p. 72). A novel way to approach this is through art in outdoor places [20,21]. There is evidence that arts education can aid physical, cognitive, linguistic, social and emotional development [22], as well as improving both mental health and social inclusion [23]. However, in 'Global North' settings, and in cities everywhere, individuals with low socioeconomic status have less access to the arts than their more affluent counterparts and the arts are increasingly marginalised in school curricula [24–26].

This paper reports on a study situated at the intersection of these three issues: a concern with children's wellbeing; their apparent disconnect with the natural environment; and a lack of engagement with the arts in school curricula—all within the context of low socioeconomic status (and, thereby, relative inequality and disadvantage) in England. The study builds on Amartya Sen's work on human capabilities as a proxy for wellbeing, developing the notion of *eco-capabilities* to explore the impact of arts in nature practice on children's wellbeing.

## 2. Literature Review

### 2.1. A Capabilities Approach to Wellbeing

Human capabilities theory originates in welfare economics [27,28] and describes a person's capabilities to access each of the 'functionings' of a human 'being', such as being: healthy; able to live with others; able to reason; able to participate in political debate and so forth. Amartya Sen describes human capabilities as a "a person's ability to do valuable acts or reach valuable states of being" [27] (p. 30), a broad range of human *functionings* that go beyond the notion of subjective and economic wellbeing. Capabilities are future oriented, aiming to provide humans with real opportunities to achieve a state of physical, emotional, intellectual and existential wellbeing in life [29], depending on the individual's personal assessment of what they value [30].

The capabilities approach looks at individuals not in terms of actual contribution or achievement (for example, to economic growth), but rather their potential [31]. Sen's theory is the starting point for the human development approach: the idea that the purpose of development is to improve human lives by expanding the range of things that a person can be and do, such as to be healthy and well nourished, to be knowledgeable, and to participate in community life. Thus, human development becomes the process of enlarging a person's capabilities to function, the range of things that a person could do and be

in her life, expressed as expanding choices [32]. Boni and Walker [33] (p. 3) purport capabilities as providing 'the freedom to enjoy valuable functionings', or the freedom to make choices in life. This is analogous with the notion of agency, as it is used in sociology, alongside structures by a range of writers including Anthony Giddens and Pierre Bourdieu, but specifically in relation to schools by Wilmott [34]. Enabling this freedom is the 'empowerment' dimension of human development [35] (p. 2) that enables us to act within and through the structures that enfold them, and it is here that education has a key role.

There is a longstanding debate in the capabilities literature about the totalising impact of specifying a list of capabilities, questioning whether there is a universally applicable and identifiable list of capabilities that all individuals have the potential to access, regardless of circumstance. Sen argues for the importance of public participation and dialogue in arriving at valued capabilities for each situation and context [36–38], leaving his framework deliberately vague to allow communities and individuals to decide what capabilities count as valuable. Nussbaum [39], on the other hand, argues the case for a universal, cross-cultural list of central capabilities for human flourishing, even one that is provisional and open to debate, and identifies ten central human capabilities that would need to be present for a fully good human life. Others develop this further, by creating a capabilities index specifically for children, pointing out that a person's capabilities may be compromised by decisions made on behalf of that person, specifically, young children (e.g., [40]). Walshe et al. [41] identify a list of 11 capabilities emerging from literature: senses, imagination and thought; autonomy; affiliation; emotions; mental wellbeing; religion and identity; play; bodily integrity and safety; bodily health; life; and other species (after Nussbaum [39]; Biggeri [42]; Di Tommaso [43]; Addabbo, Tommaso and Facchinetti [44]; Sen [27]).

Sen suggests that education ought to enhance freedom, agency and wellbeing by "making one's life richer with the opportunity of reflective choice", enhancing "the ability of people to help themselves and to influence the world" [37] (p. 18). He argues that the capability approach is based on human agency, meaning that a person is responsible for their own life and goals that matter to them; in this way, the process of identifying capabilities should entail some form of participatory and inclusive dialogue, however conceptualised [45]. This is significant within a context in which agency has widely been acknowledged as being important to wellbeing (e.g., [46]), particularly the wellbeing of children [47]. The belief that capabilities influence wellbeing has been tested in a number of studies: for example, Van Ootegem and Verhofstadt [48] found capabilities to be a successful alternative measure for wellbeing (using life satisfaction as an interpretation of wellbeing); and Muffels and Headey [49] suggest that both subjective and objective wellbeing are the outcome of the interaction process between capabilities and choices. However, Irvine et al. [50] propose that childhood in English villages is enclosed, with children's choices being limited by social structures, such as transport systems, safety concerns and so on; amongst other things, these structures limit children's agency to choose to play freely in the outdoors. In this way, we suggest that this enclosure has a detrimental effect on children's wellbeing; when coupled with the diminishing access to the arts, this, then, has the potential to further entrench and deepen the wellbeing crisis.

### 2.2. Capabilities and Relationships with/of Nature

Walshe et al. [41] suggest that there is a strong synergy between what the Sustainable Development Goals and the capability approach are trying to achieve. Sen's capability approach holds that the natural world is important to human wellbeing [51]; although this is significant, it attributes primarily instrumental value to nature, considering how it can be of benefit to people centred development, rather than having its own intrinsic value. However, the literature on this is not consistent; for example, Nussbaum's [39] central capabilities include respect for other species and nature, whilst Ballet, Koffi, and Pelenc [52] suggest that human wellbeing should be considered equal to the preservation of natural resources.

This latter position is an invitation for thinking with a posthuman perspective about the role of wild, natural and outdoor places in the establishment and maintenance of positive subjective wellbeing. If we consider people to be a part of nature [53–55] (as well as being apart from it, as has been argued elsewhere, e.g., by Bonnett [56] and Lee et al. [57]), entangled and implicit in its fate, then we have to accept that a threat to nature (such as climate breakdown) is also a threat to humanity. Such an existential issue is likely to have a very significant impact on our mental health and wellbeing. From this point of inherence, it becomes necessary for us to think about our wellbeing as entwined with that of the rest of the biosphere, and our fate as implicit in the fate of the biosphere [39]; it could be argued that the increase in eco-anxiety further exemplifies this inextricable link (e.g., [9]).

Bonnet [56] describes the exceptionalism of humanity from nature, a separation that many posit to be the result of a process of indoorisation and that has happened over the past few decades. This is explored by a number of authors; for example, the Natural Childhood Report [58] found that, in the last 30 years, the number of children regularly playing in wild places in the UK fell by 90%. There is widespread agreement that substantial benefits for wellbeing may be derived from contact with nature [59–62]; furthermore, studies of the impact of COVID-19 on children's mental health have found that regular time outdoors is associated with better mental health, whereas increases in daily screen time and less time outdoors were associated with poorer mental health (e.g., [13,14]).

Although natural environments benefit everyone, disadvantaged groups benefit the most as socioeconomic health inequities are lower in greener communities [63]. In the least green areas, mortality rates are estimated to be 93% higher for deprived groups, compared to 43% in greener areas, suggesting that greenspace weakens the effects of deprivation on health [64]. Despite this, socioeconomically deprived areas have significantly less good quality public greenspace [65], while children living in these areas are nine times less likely to have access to greenspace [66]. This unequal provision means that children who are already at risk of poor health have the least opportunity to reap the health benefits of greenspace [67].

### 2.3. The Arts as a Mechanism for Developing Capabilities and Wellbeing

There is evidence that arts education can improve both wellbeing and social inclusion (e.g., [68,69]), as well as developing children's capabilities [70]. Arts based approaches have been found to support the development of the qualities on Nussbaum's [39] list of capabilities: for example, arts performance through kinaesthetic forms such as dance and theatre have a demonstrated impact on bodily health and bodily integrity [71]; while engagement with visual arts has been shown to improve imagination and thought [72] and emotional skills (e.g., [73,74]). Despite this compelling evidence, individuals with low socioeconomic status continue to have less access to the arts than their more affluent counterparts [74] and the arts are increasingly marginalised in school curricula [26]. Thinking about this from the point of view of structure and agency, the structural imposition of the impoverishment of the arts through state policy and socioeconomic inequality diminishes the agency of these children to develop their capabilities, and this is evidenced by comparison with children who do not suffer the same structural inequalities. In this way, the arts become a form of expression that enables young people to learn to communicate their agency; by diminishing their access to them, we limit the growth of their agency, which has a concomitant impact on their mental health.

Of note is a recent systematic review by Moula, Palmer and Walshe [75,76], which synthesised existing evidence concerning the interconnectedness between arts and nature, and their impact on the health and wellbeing of children and young people. The review included eleven studies, encompassing data from 602 participants in total, and suggested that arts can offer an inclusive medium to engage all children and young people, especially those who might otherwise remain disinterested about environmental issues and disengaged with educational programs. Engagement with arts in nature was found to increase nature connectivity and make the relationship with nature explicit, as children and young

people gradually perceived themselves as part of the environment, and the environment as part of themselves. The arts based activities, which included visual arts, music-making, drama and movement, also provided creative stimuli to better understand environmental issues and to explore ways to prevent future environmental disasters, leading to greater consideration for the environment, and a potential decrease in eco-anxiety.

In response, the overarching aim of this study was to explore how working with artists in nature influences children's wellbeing. We considered the work of arts based charity Cambridge Curiosity and Imagination (CCI), which aims to create opportunities for children's creative adventures in local, familiar, outdoor places, empowering young people (and others in their wider community) with the agency to act in relation to the spaces that matter to them. In particular, research questions informing this element of project design were:

RQ1: How does working with artists in nature support children's eco-capabilities? In what ways does this impact their wellbeing?

RQ2: What are the processes by which any potential change takes place?

RQ3: What are the implications for environmental and sustainability education?

## 3. Research Design

### 3.1. Context

Four classes of pupils in Years 3–5 (aged 7–10) from two primary schools in eastern England participated in this project; a total of 101 children. Both schools were located in areas with an IDACI (Income Deprivation Affecting Children Index—which measures the proportion of all children aged 0 to 15 living in income deprived families) of fourth quintile. Over 40% of children in both schools have registered for free school meals (the national average in 2018 was 13.6%); both have above average percentages of children with special educational needs (SEN) and English as an additional language (EAL). Thus, the research deliberately focused on children living in areas of high deprivation. In terms of spatial setting, the first school was located within an inner city urban area; it had a small field surrounded by trees as its playground, as well as an enclosed 'nature area' for curriculum use and a small allotment area. This school was located adjacent to a small, public woodland area. The second school was located within a suburban setting; it had a field surrounded by trees as part of its playground, as well as access to a second field area separated from the main school grounds by a large fence.

Research was undertaken from April to July 2021, almost immediately following the third COVID-19 'lockdown' in England. As such, they had experienced an extended period of learning from home away from the school environment, and had had minimal interaction with either their teacher or peers immediately prior to joining the project.

### 3.2. The Practice of Arts in Nature

The practice of CCI artists is described as artscaping and has three key characteristics: to affect and be affected by arts, nature, place and space; to create a response from materials and feelings to express new ideas; and to enhance the environment in ways that delight [41]. CCI describe their practice through five threads [77]:

- Young people and artists building inclusive communities of collaboration and companionship;
- Being generous with space, time, materials and attention;
- Recognising the importance of slowliness (defined as a commitment to making time for creative practices and children's thinking to be fully explored and noticed [78]) and the emotional dimension of learning;
- Inviting exploration through movement, ideas, art and nature; and
- Reimagining the familiar with powerful imaginations and fantastical possibilities.

Throughout this project, artists spent eight full days with children across eight consecutive weeks. Each day comprised different activities (some outcomes of which are

illustrated in Figure 1), but generally immediately following morning registration children went outside with artists for the morning (usually returning to a designated area for their morning break). This frequently started with an activity designed to engage them with the outdoor space, such as lying on the ground with their eyes closed, drawing what they can hear, or collecting different sizes and shapes of leaves. The children then moved onto a more focused activity, such as creating land art (as illustrated in Figure 1a), sunlight photography (Figure 1e) or painting aspects of the environment (Figure 1c). During the afternoon, children often worked in or close to the classroom using a wider range of materials, such as paints (water-colour or poster: Figure 1b), chalks, oil pastels, clay, ink made from blackberries and even painting on their hands. These sessions allowed children to build on activities from the morning, reflecting on the colours, textures, patterns and environments they had explored. In addition to the eight project days, a community event was held as a ninth artist led day, comprising opportunities to share and celebrate the practice and created artwork with the wider community within and beyond the school; this entailed parents, caregivers and grandparents in one school, but in the other the community days were limited to inviting the children from the rest of the school to view the work (because of the restrictions of COVID-19). Both schools were ultimately presented with a Fantastical Map of the project (e.g., Figure 2), with all children given a smaller leaflet containing both the map and a description of the project to keep.

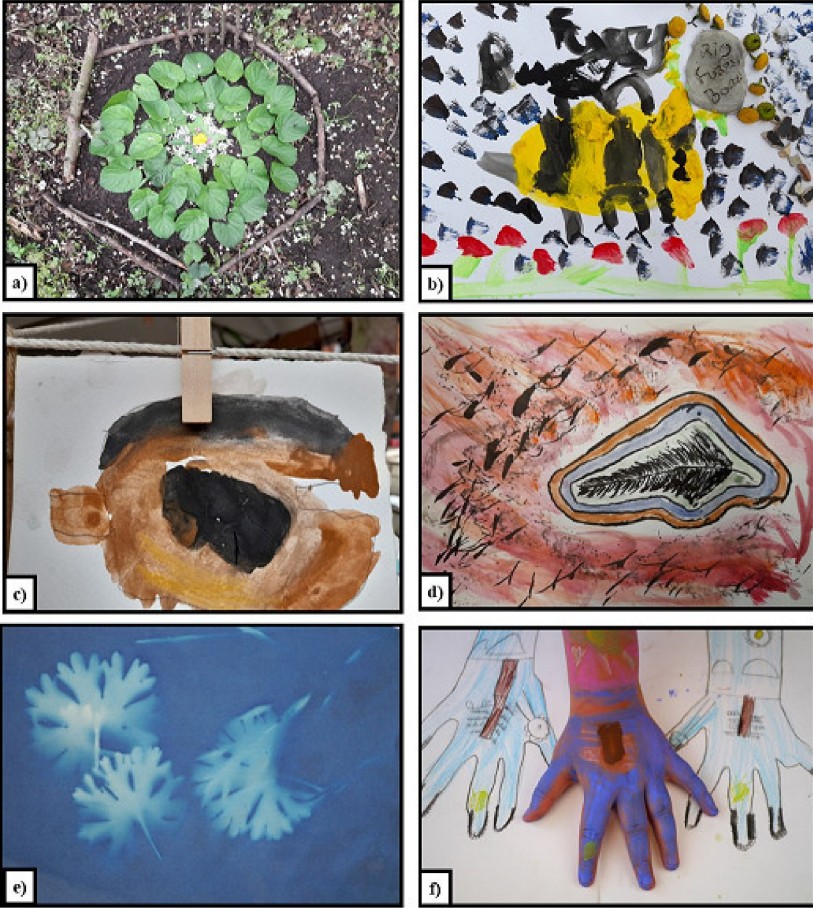

**Figure 1.** Examples of artwork produced by the children throughout the artist led days, including (**a**) land art; (**b**) post paint and clay representation of a bee; (**c**) painting of the eye of a piece of wood; (**d**) a water colour and blackberry ink painting of a feature; (**e**) sunlight photography; and (**f**) body art. Photographs (**a**–**d**) copyright Nicola Walshe; photographs (**e,f**) copyright Caroline Wendling.

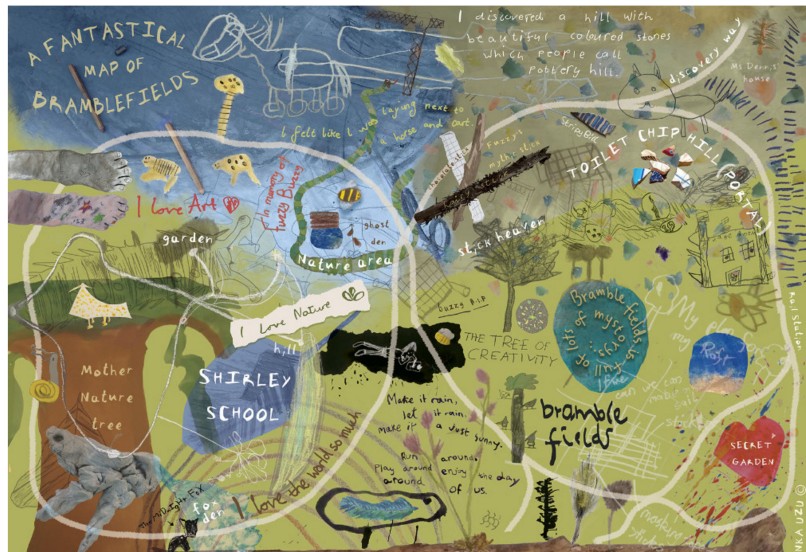

**Figure 2.** Fantastical Map of Bramblefields by Tonka Uzu with images and words from children involved in the Eco-Capabilities project at one school. Copyright Tonka Uzu.

*3.3. Methods of Data Collection*

The research involved investigating the impact of the eight week arts in nature intervention as described above, using participatory and arts based approaches. Participatory and arts based methodological approaches were used because of their potential to stimulate visual, rather than linguistic thinking [79], thereby allowing children space and time to uncover thoughts or experiences that they might struggle to verbalise [80]. In cases when words are inadequate to describe feelings and thoughts [81,82], arts based methods became an alternative way of representing children's experiences, ensuring that children's perspectives are being acknowledged and heard [83].

Before and after the artist led days (pre and post), the research team delivered a workshop that consisted of three key activities:

(a) Drawings of children's happy place: Children were invited to imagine a place where they feel happy, and draw how this place looks. They were also invited to draw five things that are important to them and they would want in their happy place, and five things that they would rather keep away from it.

(b) Walk and talk focus group: Children were invited to walk around the school grounds, to show the spaces that they like the most and the least, and to express the reasons why.

(c) Wellbeing questionnaire: The Personal Wellbeing Index—School Children (PWI—SC; [84]) was used to measure changes in children's wellbeing before and after the sessions.

During the artist led days, the research team adopted the following methods:

(a) Participant observations and fieldnotes: The researchers were participant observers in all the sessions, keeping notes about interesting behaviours, interactions, changes or 'lightbulb' moments. This also included capturing quotes from children as the activities took place. Three researchers took part in this activity, as well as artists and teachers.

(b) Reflective sessions: At the end of each day, artists, teachers and researchers gathered to reflect on their observations and any noticeable changes. These sessions lasted between 30–60 min and were audio recorded and then transcribed.

(c) Creative diaries: Children's diaries were also included for any noticeable changes in children's expressive artmaking (see examples of these in Figure 1).

Finally, at the end of the intervention, all artists, teachers and head teachers were interviewed, followed by a focus group with all artists and teachers two months later.

For the purposes of this paper, with a focus on development of children's eco-capabilities, we will incorporate data from the participant observations and fieldnotes, end of day reflective sessions, and interviews with artists, teachers and head teachers. Remaining data will be explored separately elsewhere.

### 3.4. Methods of Data Analysis

As this was a collaborative, ethnographically informed study, our process of analysis was entangled with our daily involvement with the project. Analysis in ethnography proceeds throughout, and cannot comfortably be bracketed off from any of the stages of the fieldwork; as such the ongoing conversations between ourselves, the artists, the teachers and the children were analytical and this is likely to have influenced the conclusions we draw from this data. These processes cannot be accurately captured in a linear fashion for the sake of a research article, but they are important to mention. However, alongside this ongoing process, we also set aside a specific period to focus on the analysis of our data. During this phase, all fieldnotes, reflection sessions and interviews with artists, teachers and head teachers were transcribed using the Otter.ai software and analysed using NVivo. Deductive, *a priori* analysis was undertaken using the predetermined list of eco-capabilities from Walshe et al. [41]: affiliation; autonomy; bodily health; bodily integrity and safety; emotions; identity; life; mental wellbeing; other species; play; senses, imagination and thought; and spirituality. Analysis was undertaken by two researchers individually (NW, ZM), with a review session with the third researcher (EL) to discuss and corroborate the categorization of data.

### 3.5. Ethical Considerations

This project followed BERA ethical guidelines [85] and was awarded ethical approval by the University Ethics Committee, including additional approval to undertake face to face research within the peripandemic context. Artists and teachers were fully informed prior to their decision to participate through participant information sheets and consent forms which described: the purpose of the study; the potential risks and benefits; information regarding anonymity, confidentiality, data protection and storage; and the right to withdraw. Consent was obtained from parents/guardians of children in a similar way; we held information sessions to explain the research at each school and worked alongside teachers to support those for whom accessing written information would be difficult. Assent was obtained from children in workshops at the beginning of the research by their class teachers and throughout the research by the researchers. During the initial assent workshop, teachers explained the purpose of the research and what would be expected of them using a predesigned participant information sheet. Children were then invited to write their name at the bottom, if they were happy to take part in the research. Where consent/assent was not given, children were able to participate in the artscaping activities, but no data were recorded in relation to them. Given the potential for either the discussions around wellbeing, or the artscaping practice, to reveal personal or challenging emotions in children, we developed a distress protocol and risk assessment with schools and any concerns were raised with the designated safeguarding lead as per the schools' policies.

## 4. Results and Discussion

This section firstly considers how working with artists in nature supported children's wellbeing through the development of eco-capabilities (research question 1), and then takes a thematic approach to a discussion of the processes by which this takes place (research question 2), drawing out implications for practice (research question 3).

### 4.1. How Did Working with Artists in Nature Supported Children's Wellbeing through the Development of Eco-Capabilities?

Table 1 illustrates a taxonomy of eco-capabilities emerging from the deductive analysis of interviews, focus groups, and end of day reflections with artists, teachers and researchers.

This includes the theoretical framework developed from Walshe et al. [41] alongside the updated taxonomy, amended considering the data. In the following section we will consider evidence for each of these eco-capabilities, as well as their broader contribution to wellbeing, separately. Although quotations from children captured during the artscaping days were not included in the theoretical analysis, they will be used to support the discussion.

### 4.1.1. Autonomy

There were repeated references by teachers and artists as to the development of confidence or self-reliance in the children across the project, something that we have framed as *autonomy*, but which also signifies having ownership or agency over aspects of one's life. One artist reflected:

> *The confidence of some of them really blossomed outside . . . some of them who in the classroom don't take a prominent place, they won't lead activities, they won't necessarily suggest answers to questions. When we were doing the activities you could observe them, actually taking on a lead role in group work, or showing someone else how to do something that they knew they felt confident at . . . That definitely was a remarkable change in some of them for their wellbeing and how they felt about their self-esteem.*
>
> *(artist, interview)*

This was reflected in the way the children talked about their work; for example, one said "If I'd done this at home, I would have thought I can't do it—I can't do it. But then I do it here and I can just do it and it turns out like this!" and another "I am telling myself that I am more clever than I think".

This perceived shift in confidence seemed particularly acute for some children who found it more challenging to actively participate in traditional classroom environments; this then served to empower those children, within and beyond the artist led days:

> *Children in my class who aren't as academically astute as others, and would struggle, before would be reluctant in class to answer questions, because they were quite aware of their own ability. Whereas now I've seen them absolutely flourish . . . [Child] is putting her hand up, she's giving everything a go, she's not afraid. And she's so much more confident in herself. And then that is transferred into her work, because actually, there's now 'oh, okay, I can do this by myself, I know what I'm doing'. So I've definitely seen that confidence completely grow.*
>
> *(teacher, interview)*

This was reiterated by another artist who described the children as becoming 'brave':

> *For some children, I picked up that they were academically less confident. The sheer fact that for some of them something really clicked with them in terms of the visual language of expression, that they suddenly had a sense of empowerment from it, that they thought, 'wow, I can really do this, I have a facility for this, and I can express myself', it came very fluidly to them. And from that knowing that they were very confident with something that they could do, they could hold their head up and be alongside their group, or maybe even shine a bit within the group, feel more confident than someone next to them who maybe is academically more comfortable.*
>
> *(artist, interview)*

Autonomy was found to be one of the most commonly reported outcomes in arts-based interventions delivered outdoors [69,86]. Evidence from our data suggests that the development of confidence and autonomy as an eco-capability was strong, not only within but beyond the artist led days; this then went on to further influence and impact learning across the children's week. Artists created a more open and flexible structure than might be allowed within the normal classroom setting; this supported children to develop autonomy, in doing so, enabling them to explore new limits and boundaries that might ultimately impact on their sense of agency, and thereby their capacity to engage effectively with the social structures that enclose them.

**Table 1.** Taxonomy of eco-capabilities emerging from deductive analysis of interviews with teachers and artists, focus groups with artists, and end of day reflections by artists, teachers and researchers. Table 1 shows previous coding category (from Walshe et al. [39]), alongside updated taxonomy. * The category of Life was not assigned to any data and so was removed from the final taxonomy.

| Previous Category of Capabilities [39] | Eco-Capabilities | Description | Example Quote | Number of Mentions |
|---|---|---|---|---|
| Autonomy | Autonomy | To have ownership or agency over aspects of one's life. | I think what emerged was much healthier, which was that actually the children who have confidence, were able to do their thing, but some of the children who were quieter were able to borrow from that confidence and become more independent in their work. *(artist, interview)* | 60 |
| Bodily Integrity and Safety / Bodily Health | Bodily Integrity and Safety | To have good bodily health, and protection from physical risk or danger. | I literally had fights, you know, imagine fights. They were arguing, fighting, hitting each other. Well, that didn't happen at the end, towards the end, there was nothing like that. Absolutely not, completely behaved. *(artist, interview)* | 10 |
| Affiliation | Relationality: Human | To be able to live with and toward others, to recognise and show concern for other humans, to engage in social relations, to be treated with respect and dignity. | There were a lot of things falling down a lot of the time where he got frustrated, but then other children started coming to help. And he actually said to me today, '[Miss], today is teamwork. Today is teamwork'. *(teacher, end of day reflection)* | 51 |
| Other Species | Relationality: Nonhuman | To be able to live with a concern for and in relation to animals, plants, and the world of nature. | I think they definitely started to get a real sense of how nature links with them. It's not a separate thing. It's not just a thing that's going to be there forever, just for our amusement as such. *(artist, interview)* | 123 |
| Senses, Imagination and Thought / Play | Senses and Imagination | To be able to use senses and imagination, involving freedom of expression. | They've been looking, observing, listening, you know, I think that has helped them because at a sensory level they've had that opportunity to kind of focus all of those different senses and engage with nature in a different way. *(teacher, end of day reflection)* / The wildness of Bramblefields came up and a few children said that it made them feel like it was like they were coming into their imagination. But it was real. That was nice. And when they felt like it's more adventurous. *(artist, end of day reflection)* | 82 |
| Emotions / Mental Wellbeing | Mental and Emotional Wellbeing | To love and care for those who love and care for us and to grieve at their absence. To be able to articulate/express all emotions and not have emotional development harmed by fear and/or anxiety. To be mentally healthy. | He said, if you're in a bad mood, you can go for a walk. And it takes the stress out *(teacher, end of day reflection)* / They all came together to create this memorial but then there was sort of almost acting out lots of emotions as well of crying and feeling sad about the loss of the bee. *(artist, interview)* | 63 |

**Table 1.** *Cont.*

| Previous Category of Capabilities [39] | Eco-Capabilities | Description | Example Quote | Number of Mentions |
|---|---|---|---|---|
| Religion | Spirituality | To be concerned with the spirit or soul, including but not exclusively related to, religion. | We had the burial. A little Bumblebee fuzzy . . . Such a huge ceremony, they found this little Bumblebee on the playground, decided they were going to bury it in the nature area. They created the most amazing shrine, everyone was bringing flowers. And then it turned into a whole ceremony. *(artist, end of day reflection)* | 10 |
| Identity | Identity | The qualities, beliefs, personality and expressions that characterise an individual. | The main thing that's happening here is this idea of a portrait is becoming a wider essence of representing themselves to the world. *(artist, end of day reflection)* | 11 |
| *Life \** | *Removed* | *To be able to live to the end of a human life of normal length and not having a life so reduced that it is not worth living.* | *N/A* | *N/A \** |

4.1.2. Relationality: Human

Within the structure of eco-capabilities, relatedness emerged very strongly as two distinct themes—relationality: human, and relationality: nonhuman. *Relationality: human*, for us, described the relationships that were developing over time both between children but also between children and adults (including teachers, teaching assistants and artists). This facilitated more collaborative work than teachers had observed prior to the project; within one end of day reflection a teacher articulated:

> *There's a lot of collaborative work today, very evident from outside, I've got loads of notes of different children who are working together . . . Lots of different dialogues . . . 'do you want me to help you this?' Or 'You start this and I'll do this' . . . So I just thought it today is actually like a real Wow, they're starting to come together, they're starting to think collectively rather than individually.*

<div align="right">

*(teacher, end of day reflection)*

</div>

Teachers and artists also reported children articulating their changing roles within the class throughout the project. For example, an artist recounted a child expressing 'my favourite thing about today is helping. I really like helping people'. This is particularly significant, as teachers had described children as short tempered and lacking patience with their peers following periods of isolation throughout the COVID-19 related lockdown; the project appeared to be providing a space within which children could reconnect with each other, as well as the school, constructively and safely. This then impacted classroom relationships and ethos across the rest of the week, with children being more respectful and empathetic of one another: 'Today when they were sharing [their artwork] . . . [child] was really serious and nodding, 'That's beautiful. Yeah, that's amazing'. I really noticed these genuine moments of appreciation' (teacher, end of day reflection). While this was less explicitly articulated by the children, they did sense a more collaborative way of working with each other; for example, following an activity in which all children drew together on one piece of paper one said 'We did it together with friends—he came into my drawing'.

Teachers also articulated their sense of connecting differently with children through the project:

> *I've had a lot more opportunities to just sit and talk and get to know my children more . . . especially those children who are quiet . . . I've got to know my children more and have more meaningful discussions with them rather than just 'yeah, that's okay, tell me at break time or lunchtime' . . . It's like it's breathing some space into your relationship with your children.*

<div align="right">

*(teacher, end of day reflection)*

</div>

As such, both children and adults felt a sense of increased collaboration but also an understanding, compassion and respect for one another that flowed into their wider relationships. Seligman and Csikszentmihalyi [87] argue that positive relationships and engagement with others can lead to positive emotions and an enhanced sense of meaning in life—and vice versa. When children can relate to others and develop healthy relationships, they feel more confident in their ability to fulfil their own potential (self-actualisation: [88]). Through this sense of belonging and relatedness, children can progress towards the development of autonomy, agency and mastery over their own lives [88]; all of which are enablers of children's wellbeing. The sense of working together also engenders co-responsibility, collective responsibility which, Makrakis and Kostoulas-Makrakis argue, might support them towards building a more sustainable future [89].

4.1.3. Bodily Integrity and Safety

The eco-capability *bodily integrity and safety* is defined as having good bodily health, and protection from physical risk or danger. Through its most literal interpretation, artists and teachers described the changes in the physicality of children's interactions with one another, which influenced their physical safety. While the impact of the project was less

literal in terms of bodily safety, artists and teachers frequently expressed shifts in comfort that enabled children to inhabit spaces more safely and fluidly. Artists drew on symbols of oppression to describe the shift from the classroom to the outdoors: 'I do think it's really nice for them being outside, because all of a sudden there are no walls' (artist, interview), and another reflected on the movement of children beyond a gate in the school field through which some sessions took place: 'When they got to the gate it was like they were in jail. Like the bars. That real sense of urgency, 'I need to get out,' was amazing. The anticipation, the freedom of being outside again' (artist, interview). In this way, they were engendering a sense of freedom and emancipation through their practice that, in turn, provided them more freedom to initiate their own learning experiences and to show independence of thinking and action [90]. Once in the spaces, working with artists allowed them to inhibit those spaces in different ways, described by one as a 'coming into the body, and drawing on a more kinaesthetic approach' (artist, interview). This had the impact of fostering a sense of calm that allowed them to more comfortably inhabit the spaces within which they were working, both as individuals and a community:

> *They really took command of this space. This was their studio, the canopy in the woods. And they settled very quickly, they had a real sense of ownership of the space, and excitement, but sophistication as well.*

> *(artist, interview)*

Through inhabiting these spaces more comfortably, the artists described a changing relationship with their own bodies as they became more aware of how to be comfortable within them: 'I think they are also owning their bodies a little bit more: 'I'm relaxing, I need to lie down. I'm uncomfortable' I think that's really great for kids to own their body' (artist, interview). This was also identified by the children; for example, one commented 'A lot of people find art relaxing, you get to be really creative, it gives you more freedom which relaxed you'. This process was articulated further by the artists:

> *When a child is able to be in nature, and that child is 100% absolutely tuned in, there are lots of things happening. So that child might be in conversations with a child, and in conversation with an adult, very focused working on their sketchbook or just looking at nature, in a particular detailed way. That child is showing a physical effort, you can tell their body is relaxed, the field is safe, and you feel they are really connected to the world. Enchanted, because they're connected, because they're safe. Because they are connected to their own selves. They are thriving. So they can find things within themselves, it makes them strong as little as human beings can be.*

> *(artist, interview)*

In contrast to more traditional classroom practice where children are controlled and constrained by spaces and rules (the social structures) that determine how they use their bodies, this freedom of movement and material engagement provides a more child friendly structure to support their development as agents of their own bodies. This, in turn, affects the agency they have with which to engage with cultural rules or schemas within school (as defined by Giddens, for example [91,92]) and their capabilities, as we describe them here.

### 4.1.4. Identity

Identity as an eco-capability is defined as the qualities, beliefs, personality and expressions that characterise an individual. Throughout the project, and becoming more comfortable with their physical presence within the outdoor spaces, children began to explore and develop their broader identity, both individually and collectively. For example, in one school children were led through a process of body mapping, a life size representation of self, using art, slogans and symbolism to articulate one's life story [93] (Figure 1f). This was articulated not only as children gaining a stronger sense of their own identity, but also a sense of security in being themselves: 'especially for the children who are struggling to maintain classroom etiquette, that ability to feel free but safe, to feel independent but heard. To feel that they can

be who they are but know that there's more they can become.' (artist, interview). Another artist described this as 'enjoying and being an individual' (artist, interview).

One particularly strong illustration of this is in relation to a child with traveller heritage within one school: Bartley (pseudonym). For the first three weeks Bartley engaged with physical activities but was reluctant to make any type of marks on paper. However, in the fourth session, when asked to draw something that represented their identity, he drew his horse, Patch: 'He was drawing his horse. And it's just the way he [announced] it with such pride. I mean, I already knew his struggle, but I hadn't said anything . . . It felt like quite a big moment.' (artist, end of day reflection). This was significant as he had not previously talked about his identity as a traveller within school, and yet, through working with artists, Bartley had been able to develop the confidence to share, and be proud of, an extremely important part of his traveller heritage and identity: his horse. Perhaps even more significantly, Bartley arrived at school riding Patch the following week. This suggests that restructuring the rules of the classroom space provided more agency for the children to determine how it is used and allowed them to bring their own identity into play. This is important, as research with Gypsy, Roma and Traveller children suggests a pupil's sense of belonging plays an important part in the way they engage with and experience education; pupils who feel a sense of belonging are more likely to remain in the education system [94].

### 4.1.5. Relationality: Nonhuman

*Relationality: Nonhuman* is defined as being able to live with a concern for and in relation to animals, plants, and the world of nature. Throughout the project, children were encouraged to reflect on their identity as individuals, but also in relation to nature:

> *Working with them to develop their portraits, alongside working outside with different stimuli from nature and the outside world and fusing that with ideas of your own sense of self and your own being, and how they can be linked and how they can feed each other.*
>
> *(artist, interview)*

This began with an embodied, or visceral, response to being outside (articulated as a 'kind of hunger' by one artist), and developed to a deeper noticing:

> *[Child] picked up a seed . . . He didn't know what it was. He opened it up and he said, 'Oh, there's the petals in the middle' . . . And then he said to me 'It is just like a human being. It's like a baby in the mommy's tummy. And the tummy is protecting it and the leaves are protected. And then when it's ready, it opens up and comes out.' Wow. That to me was love. Wow.*
>
> *(artist, interview)*

However, very quickly, teachers articulated that this noticing extended into and informed the rest of the school week, for example during a science lesson:

> *When we were doing identification of [insects], they were kind of linking it with the creatures that they had seen already outside doing the art . . . They didn't realise they were actually learning anything by noticing these insects. And then they were clearly observing them, and noticing features that they had. And they found that really easy, because they'd already closely observed these creatures in real life.*
>
> *(teacher, interview)*

Children also began to notice nature in their own home contexts, and were excited each week to discuss about the moth in their kitchen, or the ant on their floor. For example, at the start of one session they collectively shared 'This morning I saw a moth . . . on my way to school I saw fuzzy buzzy's brother . . . I saw a red butterfly'. As they developed a more meaningful connection with the natural environment, they began to demonstrate a keenness to protect it, taking ownership of the outdoor spaces they were visiting, thereby becoming protective of them and behaving in a more environmentally sustainable way:

*Over time, it was almost like they took ownership of that area. For real, this is our space . . . At one point, some people had trespassed and left the remains of a party, and the children were really annoyed. They were . . . very much like, 'Who are these people that have come in? How have they gotten in? What are they doing leaving all this rubbish in our area?' There was like a real sense of, it's like they'd gone into their garden. They really didn't like it.*

<div align="right">(artist, interview)</div>

Over time, this was articulated by the children throughout the artist led days; for example, one reminded the class 'Remember! We are nature! So we don't destroy it, we take care of it'. In the final session, another commented 'The art project links to climate change as it gives people imagination to do things about it'.

Previous studies have demonstrated how human–nature connectedness has a restorative effect for children's wellbeing (e.g., [68,95,96]). Ulrich's psycho-evolutionary theory [97], describes humans' innate affiliation with natural environments, drawing upon the assumption that natural environments induce positive emotions and feelings [98]. This human–nature connectedness can help people to view themselves as part of a wider ecology, which has a positive impact on aspects of wellbeing, such as vitality, creativity and happiness [99]. For some of our children, this more explicitly became empowering, a sense of being able to make a difference and care for nature; for example, during one session children found a dead bumble bee and spent a morning building a shrine and holding a burial:

*All of this focus of their emotions on this little bee was quite amazing . . . I think maybe [they felt] some empowerment from what they could do to save the bee, and fundraise for the bee, and look after the bee. Sometimes children, they're connected to climate crisis. But that can feel so huge and overwhelming. And to be able to put it in into this little bee was just amazing.*

<div align="right">(artist, interview)</div>

Another group of children who had spent their initial days in nature breaking branches and pulling at flowers and wildlife began to care for a 'family' of snails over the course of several weeks (beyond the artscaping days).

The above echo the findings of a current systematic review [76] where relatedness and connectedness to nature was achieved in all included studies regardless of the type of arts based interventions delivered. As suggested by Lumber, Richardson and Sheffield [100], relatedness emerged from being exposed to the beauty of nature, the emotions that arise while being in nature, and with sustained contact.

### 4.1.6. Senses and Imagination

The eco-capability *senses and imagination* is defined as being able to use senses and imagination with the involvement of freedom of expression. The artists articulated the embodied, sentient nature of their practice as, by way of the senses, they provided opportunities for children to engage with familiar local spaces engendering noncognitive, affective feelings and emotions:

*Using a lot of sensory prompts . . . touching the grass, maybe listening, closing your eyes, closing your eyes was possibly the most profound one . . . Some spoke about what they had imagined during that time. And some just talked about that effect of opening new eyes and you see black at first and then you see the world again . . . They become much more sort of enhanced like in a sense of smell and hearing.*

<div align="right">(artist, end of day reflection)</div>

Another artist observed 'They've moved into a more tactile way of experiencing nature' (artist, end of day reflection). This was recognized by some of the children, for example, one commented 'I feel freer . . . I love nature . . . It is beautiful so I am happy that I am nature . . . it is wild—it is good to be able to be wild . . . You can feel the air going into your skin and it

is wonderful'. As articulated in the eco-capabilities of relationality (human and nonhuman), this embodied, sentient approach appeared to facilitate children's meaningful engagement with their environment, each other and their nonhuman worlds. Rodaway [101] (p. 4) emphasises the role of the senses in "*geographical understanding, the senses both as a relationship to the world and the senses as themselves a kind of structuring of space and defining place*" (italics in original). Similarly, through 'ecologies of place', Thrift [102] stresses the significance and richness of human interaction with place (human and nonhuman), suggesting that, through taking seriously physical movement or practice in natural environments, we can better conceptualise embodied experience that is situated in time, place and space [103].

### 4.1.7. Spirituality

Spirituality as an eco-capability is defined as being concerned with the spirit or soul, including, but not exclusively related to, religion. Although a less significant eco-capability, as illustrated through the data, on occasion the artists' practice did provoke spiritual references. For example, at the end of one day an artist reflected: 'When they got through the gate, two groups said, 'We are in heaven. This is heaven''.

However, children demonstrated a broader sense of spirituality; this is illustrated through the discovery and subsequent elaborate burial of Fuzzy Buzzy, the bumble bee—as described above and articulated by one artist:

*We had the burial. A little Bumblebee fuzzy . . . Such a huge ceremony, they found this little Bumblebee on the playground, decided they were going to bury it in the nature area. They created the most amazing shrine, everyone was bringing flowers. And then it turned into a whole ceremony.*

*(artist, end of day reflection)*

Children also articulated a sense of tranquillity that allowed them to connect with their environment, for example, one remarked 'It's my first time where I've actually been quiet, 'cause everything in my house is always so loud and people shout all the time'. The concept of spirituality has been explored in a similar intervention [87]. The authors found that music-making outdoors provoked a sense of interconnectivity and harmony with nature, leading to experiences of extraordinary and transcendent moments [104,105]. Children expressed that they experienced a sense of entering a heightened reality of bonding with nature and with each other. These findings echo previous studies suggesting that it is possible for spiritual experiences to be achieved in childhood as they do not require high cognitive abilities or sophisticated language capacity, but universal human awareness [106]. Arts can be the catalyst to inspire these spiritual experiences, thus reinforcing connectedness with the environment.

### 4.1.8. Mental and Emotional Wellbeing

The final eco-capability is *mental and emotional wellbeing*, defined as to love and care for those who love and care for us and to grieve at their absence, to be able to articulate/express all emotions and not have emotional development harmed by fear and/or anxiety, and to be mentally healthy. Throughout the project there was a sense that children were becoming happy or well, both from artists and teachers; for example, one artist reflected on changes in the way that a teaching assistant worked with a child who had not always been permitted to join the class because of a risk of running away:

*The TA who was one-to-one with one of the high-needs children, when she came along to one of the visits, she didn't even follow him around all the time, which she used to do. And she just sat and she started drawing. And someone asked her, 'oh, you do not need to watch [child]?' And she said, 'No, I can just see how happy he is'.*

*(artist, end of day reflection)*

One child articulated this themselves, saying during an artscaping day 'I like painting because it calms me down'. This sense of children being happy was also noted by parents;

for example one artist remembered parents approaching her at the end of the day: 'They were just commenting how much it had transformed the children, how much the children are happy' (artist, interview).

Children also became increasingly willing to articulate their broader emotions throughout the project, as exemplified by the end of day reflections of one artist:

*I asked [child] 'do you like painting'? And she said, 'yes, because you can draw out your fears and express yourself'. I asked her, 'what do you mean by that?' And she said, 'sometimes you can't express a fear from your mouth, but you can always draw it'.*

On several days the activities provided a space to explore departed family or friends. For example:

*When we were in the garden one of them remembered somebody, somebody said, 'Oh, I miss someone and then everybody was finding somebody that they missed, someone that died or that moved away. It was a big conversation just when we were drawing the dandelions.*

*(artist, end of day reflection)*

*One child was doing a personal representation of a colour and chose these two colours. One was his grandmother's favourite colour, and one was his, and he was kind of blending them together. And it just made this whole page and he's quite careful normally with his mark making, you know, quite conscious of not spreading, and this was fantastic kind of explosion of colour for a person who has died. And it's very much about her, you know, felt like a memorial kind of thing. And he really openly shared that.*

*(artist, end of day reflection)*

In contrast to adults, children do not usually have the words to verbalise their grief and mourn. Creative activities, such as the rituals described in the previous eco-capability (i.e., spirituality), can provide a safe avenue for children to represent their feelings and honour the person who died. This became apparent during several of the sessions focusing on colour; for example, one child explained their colours: "Black means sad and angry, and green means perfect calmness". Normalising conversations around life and death through the arts and validating children's feelings can offer a sense of control, as children make sense of their emotions, and a sense of empowerment that allows them to cope better with grief, thus supporting children's wellbeing [107].

In the recent systematic review [76], all included studies identified positive changes for children's mental and emotional wellbeing; in particular, the appreciation of beauty and the unexpectedness of what can be found outdoors, such as natural sounds and sceneries, had a positive impact on children's wellbeing. This evidence can be linked both with Wilson's concept of biophilia [108], as well as Ulrich's psycho-evolutionary theory [97], according to which, all humans have an innate affiliation with natural environments and, as such, natural environments can induce positive emotions in all humans [109]. This human–nature connectedness can help children to view themselves as part of a wider ecology, affecting positively their vitality, happiness, and care for the environment [99].

*4.2. What Are the Processes by Which Eco-Capabilities Are Developed in Children?*

This section explores which characteristics of the artist practice contributed to the development of eco-capabilities (RQ2); interview, reflection and focus group data from artists and teachers suggest that the most important mechanisms of change appeared to be (a) time and consistency being outdoors; (b) materiality and embodiment; (c) a sense of slowliness; and (d) emotional expression.

The time that children were spending outdoors combined with the consistency of the practice (i.e., one full day each week) were valuable for the development of all eco-capabilities, but primarily for children's agency. It was noticeable that, due to the consistency of the practice, children's creativity and imagination were flourishing every week. While children were gaining more confidence in their skills, they felt more comfortable

taking risks and experimenting with new materials and techniques. Ownership of the outdoor spaces was also achieved gradually and increasingly every week, which was vital in increasing children's autonomy and agency. One child remarked 'We found tiny things we didn't know were there even though we've been in school for years'. Children further had the time to reflect on their self and identity every week, directly or indirectly. This often led to higher self-awareness, but also a sense that the environment was becoming part of children's identity, and they were becoming part of the environment—what has been defined as pro-environmental identity [110]. In turn, pro-environmental identity led to a seemingly higher awareness of environmental concerns, and suggestions of behaving in a way that protected and cared for the environment, thereby contributing, therefore, to sustainability.

Materiality and embodiment played a key role in allowing children to explore their boundaries and to feel safe, thus developing body integrity and safety. The embodied and sentient nature of the practice was also key to providing an affective learning response and connection to place and nature. For example, engagement with natural materials such as rocks, leaves, the sounds of the wind or birdsongs, maximised opportunities for children to explore their senses, hence developing their creativity and imagination. One child said "being out in nature—it felt like I was in the sky . . . it was real life but it felt like I was just a bird". This posthuman, embodied approach to pedagogy is increasingly recognised through environmental and sustainability education research (e.g., [111]) and is considered further in a separate publication about this research.

Slowliness is a term coined by artists working within CCI [77], and within the context of this research it clearly supported children to develop a range of eco-capabilities. For example, it offered children the opportunity to notice and be mindful of their surroundings, such as the shapes of the clouds or the bugs on the grass, contributing, therefore, to their nonhuman relationality. Slowliness also gave children time to connect with classmates that they might not have otherwise, thereby developing their human relationality. Finally, slowliness encouraged children to adopt coping mechanisms to reclaim bodily integrity, such as through deep breathing and mindful noticing, including within situations where this has been compromised, such as after an argument or an uncomfortable experience. One child explained 'You don't normally get the chance to go outside . . . There is much more time and you can be free'.

Emotional expression was key in this practice as arts provided a creative, nonverbal avenue for children to express their emotions, feelings and thoughts, even for uncomfortable experiences and circumstances, such as those of death and grief. For example, when describing their colour mixing, one child explained 'Purple and yellow—my grandma's favourite colour and mine—she died—we are mixing together'. In this way, the practice appeared to offer children a sense of relief and the discovery of coping mechanisms for negative emotions, such as the process of undertaking arts, spending more time outdoors, or both. In these ways, emotional expression nurtured and supported children's mental and emotional wellbeing, both within and beyond times of emotional distress; from this, we suggest arts in nature activities have the potential to be used as a tool with which to mitigate broader concerns, such as eco-anxiety. In addition, the positive emotions that arose while being in nature, such as calmness, vitality and gratitude, appeared to further increase children's relatedness with nature; one child said: 'It was like I was not a person, it was like I was the nature'. This bonding reinforced children's care for the environment, and appeared to contribute to their overall protection of it.

### 4.3. Limitations

While we argue the four pedagogical elements underpinning the arts in nature practice defined above had significant impact on the development of children's eco-capabilities and, ultimately, their wellbeing, it is important to acknowledge the journey towards this outcome was, at times, challenging for both teachers and children. For some children, the transition from a highly structured and tightly controlled traditional classroom environment to one

with more agency and flow was difficult, and they needed time to become used to this different, more liberated way of being and learning. Within one school, a small group of children appeared distracted and unfocused within the new outdoor spaces, and either did not engage or did so in a hurried and nonparticipative way; some even articulated a dislike of the arts in nature days after four weeks of the practice. However, exploring a new perspective of being and learning became familiar to the children over the course of the project; indeed, for some, the dissonance between their 'normal' school day and this new way of working in school—this struggle—appeared necessary to achieve the ultimate benefit of the work. Teachers, too, found the transition to a less controlled (and controlling) way of being with the children difficult; for example, during the first trip out of the school grounds they were visibly nervous, concerned about health and safety, risk assessments, and simply that children might disappear. While, of course, these concerns were no less important throughout the project, teachers began to trust both the children and the practice, and, in the final weeks, were visibly more comfortable as they could so clearly see the impact of the work on both the children and themselves. Ultimately, this then gave them the confidence to see how they might incorporate these pedagogies into their own practice. Within the structure of the Eco-Capabilities project, children and teachers had both the luxury of time to progress through these stages, but also expert guidance by the artists, which comes at considerable financial expense. Further research, then, is needed to consider how we might best support teachers to engage with these pedagogies and benefit from the physical experience of working with artists in schools without the luxury of such funding.

## 5. Conclusions

Findings of this study suggest that, through arts in nature practice, eight eco-capabilities are developed in children: autonomy; bodily integrity and safety; relationality: human; relationality: nonhuman; senses and imagination, mental and emotional wellbeing; spirituality; and identity. While many of these have been considered as being important capabilities previously (e.g., [27,39,42–44]), we argue that our framing of them as *eco*-capabilities is distinctive. This is because of their innate relationship with the arts in nature practice that develops them, and the more considered, caring and ultimately environmentally sustainable inclinations and behaviours they lead children towards. While we have presented them as eight distinct eco-capabilities, it is clear that they are entangled and interconnected with one another. For example, the sense of relatedness with each other can be enhanced by connectedness and relatedness with nature, as growing affection and emotional attachment to nature positively affects the connection, fondness and trust between people (e.g., [112]). This was demonstrated across the series of days with children, as they gently connected with the outdoor spaces within which they explored, in doing so developing a more collaborative, respectful and considered way of being with each other and their teacher.

The arts in nature practice appears to support the development of eco-capabilities through four main pedagogical elements: extended and repeated episodes of time undertaking arts outdoors, which allows children to feel comfortable with what become familiar unstructured routines; embodied practice that engages children affectively through the senses and helps them rediscover familiar outdoor spaces with a fresh sense of place; a sense of slowliness, which envelops children with time and space to (re)connect with the more than human world; and thoughtful practice, which facilitates and encourages emotional expression, in doing so giving children the tools with which they can develop resilience against broader worry and anxiety. We suggest that, through these four elements, the arts in nature practice guides children towards a more entangled relationship with nature, and a clearer understanding of themselves as part of it, thereby motivating them to take better care of it for themselves.

Through Eco-Capabilities, we have begun to illustrate how an artistically enriched and liberated experience of a very familiar outdoor place can provide children with the opportunities to grow and develop the kinds of capabilities that are needed for a flourishing life; this is particularly important for those children who have been structurally disadvan-

taged and, therefore, have been shown to have both greater risk of poor mental health and wellbeing, and less access to the arts and nature. By identifying the structural inequalities that inhibit this development and diminish the quality of experience of these children (e.g., lack of access to outdoor spaces, diminished arts provision, and highly controlled bodily movement within a traditional classroom), the arts in nature practice we describe here provided agentic power which enabled learning of a different kind. It enables children to be agentic despite being enclosed by social structures. While the majority of these eco-capabilities do not explicitly appear on the National Curriculum within England, their development appeared to have implications for wider curricular (and school) participation, as children developed a clearer and more assured sense of their individual and collective identity and agency. The Sustainable Development Solutions Network [113] emphasized that working with children provides a window of opportunity to lay strong foundations for long term knowledge, skills and attitudes that can play a substantial role in societal improvements. Engaging children has been also recognised as a key element in promoting a lifelong disposition towards caring for others and the environment [114]. While we suggest further, longitudinal research would be beneficial in order to more specifically consider the longer term implications on both children's wellbeing and broader learning, we anticipate the impacts of the development of eco-capabilities in children through arts in nature practice has the potential to be felt within and beyond school. Instead of more traditional, humancentric sustainability education that imposes the responsibility of saving the world on children, this could then serve to provide a more gentle and empowering way to engage children with issues of environmental sustainability.

**Author Contributions:** Authors have contributed to this paper as follows: Conceptualization, N.W. (60%), Z.M. (10%) and E.L. (30%); methodology, N.W. (45%), Z.M. (15%), and E.L. (40%); validation, N.W. (40%), Z.M. (40%), and E.L. (20%); formal analysis, N.W. (40%), Z.M. (50%), and E.L. (10%); investigation, N.W. (40%), Z.M. (45%), and E.L. (15%); data curation, N.W. (10%) and Z.M. (90%); writing—original draft preparation, N.W. (80%), Z.M. (15%), and E.L. (5%); writing—review and editing, N.W. (50%), Z.M. (40%), and E.L. (10%); supervision, N.W. (100%); project administration, N.W. (50%), Z.M. (40%), and E.L. (10%); funding acquisition, N.W. (75%) and E.L. (25%). All authors have read and agreed to the published version of the manuscript.

**Funding:** This work was funded by the Arts and Humanities Research Council under Grant AH/S006206/1.

**Institutional Review Board Statement:** The study was conducted in accordance with the Declaration of Helsinki, and approved by the Ethics Committee of Anglia Ruskin University (reference 19/20/019, 19 December 2019).

**Informed Consent Statement:** Informed consent was obtained from all subjects involved in the study.

**Data Availability Statement:** Not applicable.

**Acknowledgments:** We would like to thank the children, artists, teachers and headteachers who worked with us throughout the Eco-Capabilities project, and without whose enthusiastic participation this work would not have been possible. We are also grateful for the constructive comments that the reviewers provided on earlier drafts of this article.

**Conflicts of Interest:** The authors declare no conflict of interest. The funders had no role in the design of the study; in the collection, analyses, or interpretation of data; in the writing of the manuscript, or in the decision to publish the results.

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
