# Peer review of "Eco-Capabilities as a Pathway to Wellbeing and Sustainability"

_sustainability, doi:10.3390/su14063582_

Round 1
Reviewer 1 Report
I appreciate the opportunity to review this paper and thoroughly enjoyed reading it. In this paper, the authors describe the findings of an eight-week long arts in nature programme delivered at two primary schools in eastern England. Using deductive analysis based on Walshe et al.’s categories of capabilities, the authors present support for an adapted eco-capabilities list. They also describe the four pedagogical elements of the arts in nature programme that help to explain their findings. This work lends support to the use of arts in nature programmes at schools, particularly those with high numbers of children from low SES families, to help facilitate connection to nature and care for nature in younger children. The inclusion of various perspectives from multiple data sources and involvement of the researchers in the programme are particular strengths of this research. Additionally, given that work in this field is typically focused on the experiences of wealthier children, the involvement of children living in areas of high deprivation in this project offers an important addition to the literature.
This paper is compelling, makes a unique contribution to the literature, and would certainly be of interest to the readers of the journal. The Introduction and Literature review sections are particularly well-written. My main concerns relate to the lack of transparency about the analysis methods. Specifically, the authors note in section 3.4 that they used thematic analysis to analyse the data through a deductive approach. Later in the paper, however, the authors refer to the use of content analysis. These two analysis methods, while distinct and adhering to quite disparate approaches to qualitative work, seem to be referred to interchangeably in this paper. The authors should provide much more clarity and description of exactly what method was used (or, if multiple methods of analysis were used, this should also be made explicit) and how their analysis was carried out. If the authors have used thematic analysis for this work, the language used to describe the development of themes should be changed as, in the words of Braun and Clarke (2019; 2021), themes do not emerge.
Additionally, given the authors’ involvement in the research, a description or, at the least, acknowledgement of their positionality is needed. It is also not clear specifically which data have been included in the analysis – the authors seem to refer to seven sources of data but only state that ‘all reflections, focus groups, and interviews’ were used for analysis. It’s not immediately evident which participant voices are represented through these data sources nor why the other data sources were excluded (other than the wellbeing questionnaires).
Further detail about the context of the schools and what the arts in nature programme entailed each week (e.g., specific examples of activities, where on the school grounds this took place, what materials were used, etc.) would be helpful in providing enough information for the reader to understand if these findings are transferable to their own settings. Finally, explicit reference to both the transferability of the findings and the limitations of the study are needed. These additions will make the research, which has direct and valuable implications for practice, more rigorous and easier to understand.
I also have more specific feedback:
- Lines 32-33 need a reference (or several)
- Line 41 – some readers may not know the acronym SEN/ND, so I’d suggest writing it out in this instance
- Given that the participants are from schools in high deprivation areas, a brief discussion in the Literature Review about disparities in nature access and a tendency for research in this field to focus on wealthier children would help provide context for why this research is important and fills a gap
- Line 217 – more detail about the schools would be helpful (e.g., were they in rural, urban, or suburban areas? Did they have much green space on-site?)
- Line 256 – did research-led workshops take place every week? How did this work in the context of the child’s school day? More detail about how this programme was implemented would be useful
- Lines 304-305 – how was assent obtained from the children?
- Lines 361-363 – the inclusion of SDT here is valuable, but it should be described in the Literature Review before being referenced in the Findings/Discussion section. Additionally, when discussing SDT, the authors describe evidence of only two of the basic psychological needs but neglect to mention competence in much detail. I’d imagine that competence was certainly supported in this programme – was this not observed/was there a particular reason for not including further description of this need? Ryan and Deci emphasise the need to meet all three needs to experience wellbeing, so the inclusion of SDT feels incomplete without evidence of the third need being supported. I also note that the authors use the language of ‘agency and mastery’ rather than Ryan and Deci’s autonomy and competence – I’m unfamiliar with the use of these alternate terms. Is there a reason they opted to use those terms?
- Line 389 – this reference to teachers and artists reporting what children said made me wonder why child voices weren’t included in the analysis, particularly since it seems that data were collected from the children. Justification of why these data sources were omitted is necessary.
- Sections 4.1.3 and 4.1.4 both lack any connections to literature and thus don’t have a ‘Discussion’ element to them. These themes should be contextualised in the same way the other sections are.
- Lines 635-637 – it’s not clear how the authors developed these findings from teacher and artist data. This connects back to larger concerns about transparency regarding how analyses were conducted. If these findings were developed thematically, supporting quotations and specific observation-based examples are needed to evidence these assertions.
- Line 662 – the term ‘slowliness’ is mentioned several times throughout the paper and attributed to the CCI artists, however it’s never explained
- The assertions made in lines 678-682 feel unfounded given that the reader isn’t sure how these findings were developed.
I believe that this paper would be a valuable contribution to the journal and, indeed, to the wider field. However, considerable changes are needed to improve the transparency of the analyses presented here, amongst the other feedback noted, to increase the rigour of the work and prepare it for publication.
Author Response
Thank you to the reviewer for their very detailed and constructive comments and suggestions, we very much appreciate the time taken to provide them. We respond to them individually below (comment in black and response in purple), with changed to the manuscript highlighted using track changes
Reviewer 1:
I appreciate the opportunity to review this paper and thoroughly enjoyed reading it. In this paper, the authors describe the findings of an eight-week long arts in nature programme delivered at two primary schools in eastern England. Using deductive analysis based on Walshe et al.’s categories of capabilities, the authors present support for an adapted eco-capabilities list. They also describe the four pedagogical elements of the arts in nature programme that help to explain their findings. This work lends support to the use of arts in nature programmes at schools, particularly those with high numbers of children from low SES families, to help facilitate connection to nature and care for nature in younger children. The inclusion of various perspectives from multiple data sources and involvement of the researchers in the programme are particular strengths of this research. Additionally, given that work in this field is typically focused on the experiences of wealthier children, the involvement of children living in areas of high deprivation in this project offers an important addition to the literature.
This paper is compelling, makes a unique contribution to the literature, and would certainly be of interest to the readers of the journal. The Introduction and Literature review sections are particularly well-written.
My main concerns relate to the lack of transparency about the analysis methods. Specifically, the authors note in section 3.4 that they used thematic analysis to analyse the data through a deductive approach. Later in the paper, however, the authors refer to the use of content analysis. These two analysis methods, while distinct and adhering to quite disparate approaches to qualitative work, seem to be referred to interchangeably in this paper. The authors should provide much more clarity and description of exactly what method was used (or, if multiple methods of analysis were used, this should also be made explicit) and how their analysis was carried out. If the authors have used thematic analysis for this work, the language used to describe the development of themes should be explained.
We have edited the content to clarify this, see page 7.
Additionally, given the authors’ involvement in the research, a description or, at the least, acknowledgement of their positionality is needed.
While we are not exactly clear what this means, we assume that it is referring to the ethnographic nature of our study and so we respond in that regard. While we do not claim this to be an ethnography because we only visited the school over a period of four months, we are heavily influenced by ethnography and its assumptions about how knowledge can be unearthed, that is to say that knowing in a social sense emerges from being in the lifeworld of the individuals involved in the research. Thus, we used participant observation as a key method for gathering data. Participant observation acknowledged that the researcher has an influence on the context that is being studied, in fact the researcher comes to know about the context through their very participation in it. While this means that the outcomes of the intervention might have been somewhat different without the researchers’ presence, it does not invalidate the findings of this project. We have added a short acknowledgement of this fact on page 7.
It is also not clear specifically which data have been included in the analysis – the authors seem to refer to seven sources of data but only state that ‘all reflections, focus groups, and interviews’ were used for analysis. It’s not immediately evident which participant voices are represented through these data sources nor why the other data sources were excluded (other than the wellbeing questionnaires).
We have added a sentence to make this clear on page 7.
Further detail about the context of the schools and what the arts in nature programme entailed each week (e.g., specific examples of activities, where on the school grounds this took place, what materials were used, etc.) would be helpful in providing enough information for the reader to understand if these findings are transferable to their own settings.
We have added in further information to the Research Design, as requested (see detail below for specific information).
Finally, explicit reference to both the transferability of the findings and the limitations of the study are needed. These additions will make the research, which has direct and valuable implications for practice, more rigorous and easier to understand.
We have made more explicit where we have considered limitations and transferability of findings, to support this (see page 19: section 4.3).
I also have more specific feedback:
- Lines 32-33 need a reference (or several)
We have added in two references on page 1, lines 34.
- Line 41 – some readers may not know the acronym SEN/ND, so I’d suggest writing it out in this instance
We have written out Special Educational Needs / Neurodevelopmental differences (SEN/ND), as suggested, on page 1, line 42.
- Given that the participants are from schools in high deprivation areas, a brief discussion in the Literature Review about disparities in nature access and a tendency for research in this field to focus on wealthier children would help provide context for why this research is important and fills a gap
We have added a very brief discussion about disparities in nature access between areas of high and low deprivation on page 4.
- Line 217 – more detail about the schools would be helpful (e.g., were they in rural, urban, or suburban areas? Did they have much green space on-site?)
We have added further information about the schools on page 5 (line 235 onwards).
- Line 256 – did research-led workshops take place every week? How did this work in the context of the child’s school day? More detail about how this programme was implemented would be useful
We have added further information on the artist-led days on page 6.
- Lines 304-305 – how was assent obtained from the children?
We have added a sentence to explain how assent was obtained from children on page 8.
- Lines 361-363 – the inclusion of SDT here is valuable, but it should be described in the Literature Review before being referenced in the Findings/Discussion section. Additionally, when discussing SDT, the authors describe evidence of only two of the basic psychological needs but neglect to mention competence in much detail. I’d imagine that competence was certainly supported in this programme – was this not observed/was there a particular reason for not including further description of this need? Ryan and Deci emphasise the need to meet all three needs to experience wellbeing, so the inclusion of SDT feels incomplete without evidence of the third need being supported. I also note that the authors use the language of ‘agency and mastery’ rather than Ryan and Deci’s autonomy and competence – I’m unfamiliar with the use of these alternate terms. Is there a reason they opted to use those terms?
Thank you for drawing our attention to this. As this is not central to the paper, and in order to limit growth in the word count of the paper, we have removed this reference to Ryan and Deci SDT.
- Line 389 – this reference to teachers and artists reporting what children said made me wonder why child voices weren’t included in the analysis, particularly since it seems that data were collected from the children. Justification of why these data sources were omitted is necessary.
Thank you for this comment, we have now added in children’s voice across the results and discussion through quotations captured by participant observation throughout the artscaping days.
- Sections 4.1.3 and 4.1.4 both lack any connections to literature and thus don’t have a ‘Discussion’ element to them. These themes should be contextualised in the same way the other sections are.
We have added reference to literature within these sections, as suggested; however, this is limited to keep the length of the paper appropriate.
- Lines 635-637 – it’s not clear how the authors developed these findings from teacher and artist data. This connects back to larger concerns about transparency regarding how analyses were conducted. If these findings were developed thematically, supporting quotations and specific observation-based examples are needed to evidence these assertions.
We have added supporting quotations and observations to evidence assertions made in this section (Section 4.2).
- Line 662 – the term ‘slowliness’ is mentioned several times throughout the paper and attributed to the CCI artists, however it’s never explained
We have defined the term ‘slowliness’ on the first time of mentioning it, on page 6, line 254.
- The assertions made in lines 678-682 feel unfounded given that the reader isn’t sure how these findings were developed.
We have added in quotations to Section 4.2 to address this.
Reviewer 2 Report
This is a very interesting topic and article which is well structured and theoretically supported. Much of the literature is based on exploring the concept of capabilities and its antecedents and less to the eco-capabilities and education for sustainability. Statistical analysis was employed for the wellbeing questionnaires; however, these results were not presented because as the authors argued they were not relevant to the development of the eco-capabilities. This has to be discussed further and since a mixed-methods design was initially adopted, the impact of excluding the quantitative results should be made clear. The concept which of eudaimonic wellbeing (self-actualisation and fulfilling one’s potential) has been also brought, which originates in Aristotle. Thus, looking into an article entitled Responsibility and Co-Responsibility in Light of COVID-19 and Education for Sustainability through an Aristotelian Lens could be useful, also in getting ideas for connecting to sustainability education.
Author Response
Thank you to the reviewer for their very helpful comments and suggestions. We respond to them individually below (comment in black and response in purple), with changed to the manuscript highlighted using track changes.
Reviewer 2:
This is a very interesting topic and article which is well structured and theoretically supported. Much of the literature is based on exploring the concept of capabilities and its antecedents and less to the eco-capabilities and education for sustainability.
Statistical analysis was employed for the wellbeing questionnaires; however, these results were not presented because as the authors argued they were not relevant to the development of the eco-capabilities. This has to be discussed further and since a mixed-methods design was initially adopted, the impact of excluding the quantitative results should be made clear.
The data presented in this paper are part of a wider study which researches not only children’s eco-capabilities (the focus of this paper), but their broader wellbeing. We have described all data collected as part of the wider study for sake of completeness; however, the quantitative data does not contribute to the specific discussion on eco-capabilities so we have not included it within this paper. We have added a brief sentence to better signpost this on page 8 and removed the section on statistical analysis.
The concept which of eudaimonic wellbeing (self-actualisation and fulfilling one’s potential) has been also brought, which originates in Aristotle. Thus, looking into an article entitled Responsibility and Co-Responsibility in Light of COVID-19 and Education for Sustainability through an Aristotelian Lens could be useful, also in getting ideas for connecting to sustainability education.
Thank you very much for this suggestion, we have incorporated a reference to the Makrakis and Kostoulos-Makrakis (2021) article, as you suggest – see page 13.
Reviewer 3 Report
Additional comments or suggestions to be sent to the author (s)
In my opinion, I think that the paper is very well written. The authors have done a noble piece of work and it is very remarkable research work. In addition, the authors have taken adequate time to study the topic from the literature review on this research work.
The main objectives of the research are defined at the introduction of the study. The authors described the study problem and research questions, the importance of the study.
- The literature review covers the most important and relevant international literature sources in an appropriate structure. The literature sources are highly acceptable and most of the relevant literature sources are used, the in-text citations are used well, citation style is correct.
- All the tables and figures are clear, understandable, and relevant, sources are indicated in each cases well.
- The authors have completed the necessary evaluations. Conclusions and recommendations are well structured, those are in relevance with the analysis and discussion. Conclusions are suitable for gaining new results and initiating further or new research. The new results are drawn up in an understandable way.
- In materials and methods are a good and comprehensive overview of the topic, based on a wide range of literature. The methodological contains a correct description of the methods applied. The econometric calculations are well documented and supported.
- The conclusions show that the authors have a piece of good and deep knowledge of the topic. Novel findings and recommendations are well articulated.
Author Response
Thank you to the reviewer for taking the time to read this paper, and for the positive feedback. No changes were required by reviewer 3, so no changes were made in response to their comments.
Round 2
Reviewer 1 Report
Thank you for the opportunity to look at the revised version of this paper. The authors have addressed all comments very thoroughly, and my initial concerns have been resolved. This paper is now suitable for acceptance in the journal, and I feel that it will be of interest to many readers. Well done to the authors on a very original and well-written paper!